# Chemical, Technological, and Sensory Quality of Pasta and Bakery Products Made with the Addition of Grape Pomace Flour

**DOI:** 10.3390/foods11233812

**Published:** 2022-11-26

**Authors:** Jaqueline Menti Boff, Virgílio José Strasburg, Gabriel Tonin Ferrari, Helena de Oliveira Schmidt, Vitor Manfroi, Viviani Ruffo de Oliveira

**Affiliations:** 1Postgraduate Program in Food, Nutrition and Health, Federal University of Rio Grande do Sul (UFRGS), Porto Alegre 90035-003, RS, Brazil; 2Department of Nutrition, Federal University of Rio Grande do Sul (UFRGS), Porto Alegre 90035-003, RS, Brazil; 3Nutrition Course, Federal University of Rio Grande do Sul (UFRGS), Porto Alegre 90035-003, RS, Brazil; 4Institute of Food Science and Technology, Federal University of Rio Grande do Sul (UFRGS), Porto Alegre 90035-003, RS, Brazil

**Keywords:** *Vitis*, grape flour, grape seed, breads, baking, noodles

## Abstract

Grapes are one of the most cultivated fruits in the world. Concomitantly, a large amount of waste is generated from this product. Grape pomace (GP) flour can be used as an increment for making new food products. GP is rich in fibers and phenolic compounds, and in addition could be used to reduce agro-industrial residues. Thus, the objective of this study was to evaluate the influence of the addition of different percentages of grape pomace (GP) on the chemical, technological, and sensory characteristics in pasta and bakery products. An integrative review was conducted. The selection of papers was carried out by searching studies in databases. An improvement in chemical quality with the addition of GP was observed, such as fiber, anthocyanin, and micronutrient content. Biscuits, cookies, cakes, breads, and pasta did not show any impairment in terms of acceptability. However, biscuits and cookies had the best global acceptance. The degree of acceptance still seems to be low for the use of GP to be included in high quantities in successful products. Samples with a maximum of 10% addition of GP flour seem to be accepted. On the other hand, the higher the percentage of GP flour, the healthier benefits they seem to promote.

## 1. Introduction

Improving the nutritional quality of products that are consumed by a proportion of people is an important strategy to meet consumers who seek better nutritional options and health benefits [1]. Non-conventional flours have been studied to increase the nutritional value of traditional flours, such as wheat, and compensate for the loss of nutrients that occurs after the refining process [2], for example, a mixture of flours from other options that are considered promising in baking [3]. Currently, there are some alternatives in the use of unconventional flours in the bakery industry. Depending on the type of processing, the characteristics of these flours, and the degree of substitution, either total or partial, they can provide benefits to products with better potential in relation to those that use only conventional wheat flour [4]. Improvements in technological and nutritional characteristics have already been observed, without compromising the final acceptance of bakery products, resulting in benefits for consumer’s health [5,6].

Baked goods are usually cereal-based products, widely consumed, and highly commercialized due to their pleasant flavor and soft texture [4]. Pasta, traditionally an Italian dish, is accepted by many cultures and is available as a main source of food energy around the world [7]. In terms of nutritional value, pasta is rich in carbohydrates, but is deficient in the quantity and quality of proteins, minerals, and other nutrients [8].

Consumer health awareness leads to an increase in demand for healthier products [1]. Functional food can also be considered a promising alternative for the application of new ingredients, including from an economic, nutritional, technological, or environmental point of view [9].

Grape (*Vitis* spp.) is one of the most cultivated fruits in the world [6] and is composed of sugars, acids, pectins, gums, and aromatic compounds, in addition to phenolic compounds. Grapes and their by-products have flavonoids, the most important of them being anthocyanins, which occur naturally as glycosides dissolved in cellular fluid, particularly in the epidermal tissue. Their concentration in food is conditioned by several agronomic factors, such as cultivar, climate, soil type, and agricultural practices [10]. Anthocyanins also act as a filter of ultraviolet radiation of the leaves and, when consumed in human food, perform antioxidant activity, promoting positive impacts on consumer health [11].

Grape pomace (GP), generated mainly from wine production, is considered a by-product because there are grape skins, pulps, and seeds in pomace. The flour made from grape pomace (GP) has the potential to be used in cookies, breads, cereal bars, homemade pasta, vitamins, and juices, among other things [4]. Flour consists of dried and pulverized grape pomace. This processing allows for a longer shelf life and increases storage capacity [12]. Furthermore, the dehydration of the residue allows components of interest, such as fibers and phenolic compounds [13], to be valuable sources of nutrients [8] and could improve the nutritional value of products. Previous studies have investigated the incorporation of GP flour into fruit jams, tomato purees, and yogurts [14].

Considering the huge production of grape-based beverages, and that their grape pomace flour can be reused in the preparation of foods with modified standard formulation and could provide benefits to consumers and to the environment, this paper aimed to evaluate the influence of the addition of different percentages of grape pomace flour on the chemical, technological, and sensory characteristics of pasta and bakery products.

## 2. Materials and Methods

An integrative review was conducted. The selection of papers was carried out by searching studies in databases: Capes Periodicals, Science Direct, Lilacs and Scielo. The studies were selected using the following descriptors: ‘baking’, ‘pasta’, ‘grape flour’, ‘grape pomace’, ‘grape bagasse’, and ‘grape seed flour’, alone or in combinations, in English, Portuguese, or Spanish. The selected papers were read in their entirety, to confirm whether they covered the study subject. Data evaluated included authors, year of publication, objectives, methods, and results. The inclusion criteria were: (1) to be compatible with the main subject; (2) be available for reading. The exclusion criteria were: (1) not being compatible with the main theme; (2) study with animals; (3) be a review of literature, books, or theses.

After searching for the papers in the databases, categorizing, interpreting the results, and evaluating the studies, 24 studies were selected to integrate this review because they used flours from grape pomace in products such as breads and cakes, muffins, biscuits, cookies, and pasta. Subsequently, these remaining papers were read in their entirety, in order to confirm if they covered the subject of the current study. The data evaluated included authors, year of publication, objectives, methods, and results. These selected papers were published from 2011 onwards.

## 3. Results and Discussion

Of the 24 papers evaluated, 20.8% of the studies produced breads (*n* = 5), 4.1% made cakes (*n* = 1), 12.5% muffins (*n* = 3), 16.7% cookies (*n* = 10), 16.7% (*n* = 4) made spaghetti, fettuccini, or rigatoni, and 4.1% (*n* = 1) made noodles. Among these studies (*n* = 24), 87.5% performed chemical analyzes (*n* = 21), 79.1% performed technological analyzes (*n* = 19), and 87.5% performed sensory analysis (*n* = 21).

In Table 1 are shown the studies that evaluated pasta and bakery goods made with grape pomace, either in combination or not with other flours. Regarding the percentages of substitution of conventional flours for GP flour (Table 1), a huge variation was observed. In studies with breads, the minimum percentage found was 2% and the maximum was 15%. For cakes and muffins, the percentages were 4–20%, while for cookies and biscuits, the contents were between the values of 2–30% of minimum and maximum replacement. Pasta had the most variation in GP flour replacement, in which the minimum amount found was 1%, while the maximum was 75% of grape flour replacement.

Of the studies, 45.8% (*n* = 11) also mentioned the type of grape they had used: purple grapes—‘Merlot’, ‘Tannat’, ‘Bordeaux’, ‘Isabel’, ‘Muscatel’, ‘Pinot Noir’, and ‘Cabernet Sauvignon’; and white grapes—‘Emir’, ‘Zelen’ and ‘Riesling’.

### 3.1. Chemical Characteristics with Grape Pomace

Flours made of fruit such as grapes can be an alternative for consumption, because they are considered a source of nutrients and fibers. Fruit residues added to food products as ingredients have been associated with healthy products by consumers, because they can modify or enhance sensory and nutritional quality [36].

#### 3.1.1. Breads

Mildner-Szkudlarz et al. [15], observed that breads with the highest addition of GP flour were characterized by a significantly higher total dietary fiber level, 10% on average, when compared to the rye bread (control). They were also characterized by a significantly higher antioxidant activity, associated with their content of phenolic compounds; the increase in the concentration of grape flour caused an increase in total phenolics content. The bread with the addition of 10% grape flour presented ash content about 41% higher when compared to rye bread (control).

According to Hayta et al. [17], the total phenolic content of homemade breads (2, 5, and 10%) containing white grape pomace flour (’Emir’ variety) was investigated. The incorporation of grape pomace flour at about 5% in the bread formulation contributed positively to the total phenolic content. Although there is no agreement in the literature on the effect of baking on the antioxidant properties of breads, GP is a source of phenolic compounds, such as proanthocyanidins; furthermore, grape pomace extracts have high antioxidant and free anti-radical activity.

Sporin et al. [19] analyzed two cultivars of white and purple GP flour (‘Zelen’ and ‘Merlot’). Both GP flour cultivars showed positive correlation with phenolic content and antioxidant activity; however, phenolic contents and antioxidant activities were higher, when ‘Merlot’ GP flour was used. The same authors analyzed ‘Zelen’ and ‘Merlot’ GP flour in breads, and they observed it is an excellent source of nutritional enrichment.

Meral and Dogan [18] demonstrated that the antioxidant activities of artisan bread significantly improved with the increase in the concentration of grape seed flour. The total phenolic content increased significantly with higher levels of grape seeds in bread, with the highest content in bread being 7.5% of grape seed flour. The final antioxidant capacity of baked products can derive from the intrinsic phenolic compounds of the flour, the other ingredients that naturally contain phenolics, and the intermediate phenolic products that can be generated during cooking (for example, through Maillard reactions), among other [19]. The antioxidant activity of grape products is influenced by their total content of polyphenols and by their phenolic composition, with most red grapes having higher levels of phenolic compounds and antioxidant activity than the white varieties, due to the anthocyanin content present in the red varieties [37].

Hoye and Ross [16], in their study with breads, evaluated 2.5, 5, 7.5, and 10% of ‘Merlot’ grape seed flour, and they observed that the values of total phenolic content increased significantly according to the increase in these replacement amounts.

Incorporating dietary fiber into frequently consumed foods is an effective approach to meeting daily dietary fiber intake recommendations, often not met by most of the adult population, and meeting consumer demands for more nutritious food products [3]. Mildner-Szkudlarz et al. [15] also showed that the addition of GP significantly improved the dietary fraction content. Breads with the addition of GP were characterized by a significantly higher total dietary fiber level, 10% on average, when compared to the bread control. The interaction between gluten fibers and proteins can reduce gas retention and mass expansion, which can influence product acceptance [9].

#### 3.1.2. Cakes and Muffins

Nakov et al. [21] observed that the addition of GP flour to cakes from 4–10% increased nutrients (Table 2). Yalcin, Ozdal, and Gok [22] also found increased moisture, protein, and lipid contents in all muffin samples, according to the amount of GP flour. Ash content in muffins made with ‘Riesling’ and ‘Tannat’ grape flour were also higher when compared to the control muffins [6]. Moisture was similar in five types of muffins studied by Ortega-Heras et al. [20], in which the results were considered low, which can be explained by the fact that the grape pomace used are low hygroscopic. The decrease in moisture found in the cakes seems to be directly related to the low moisture content of the grape pomace [21]. However, the increased moisture content in the breads can be attributed to the higher water absorption of the fibers present in the grape pomace flour [15].

For Nakov et al. [21], the lower pH found in cakes enriched with GP is probably related to the presence of organic acids. ‘Tannat’ grape flour had high protein content, and these muffins were different from the control.

The composition of free phenols in the cakes showed significant differences between the control and samples enriched with ‘Muscatel’ GP flour. The total concentration of phenols in the cakes enriched with GP was significantly higher when compared to the control cake. Such results are similar to those found by Yalcin, Ozdal, and Gok [22] regarding total phenolic content and antioxidant capacity in muffins.

Bender et al. [6] observed that the dietary fiber content of the muffins increased as the amount of grape skin flour of the two varieties increased. The same could be seen in Ortega-Heras et al. [20]; the samples had higher fiber content when compared to the control.

#### 3.1.3. Biscuits and Cookies

For Abreu et al. [28], the nutritional composition of grape flour showed lower moisture and carbohydrate content and higher ash and fiber values when compared to wheat flour.

Karnopp et al. [25] noticed that the enrichment of cookies with GP promoted a significant increase in total phenolic compounds. The same occurred with the total phenolic content of the cookies made by Aksoylu, Çagindi, and Köse [38]. Cookies made with 30% GP and whole wheat flour showed greater antioxidant activity and significantly increased the fiber content. However, the authors observed water activity decreasing. Protein and lipid contents were not affected by GP and whole wheat flour [25].

By contrast, Theagarajan et al. [30] noticed that the protein content of biscuits showed a significant difference between the control and biscuit samples, with the highest content being 8%. Furthermore, there was significant difference between the ash content of the control and test biscuits, which reflects the increase in the mineral content of biscuits with incorporated pomace.

Acun and Gül [24] concluded that the total content of dietary fiber and total phenolic compounds of cookies increased as the level of GP increased, especially cookies with 15% seedless grape flour.

For Maner, Sharma, and Banerjee [26], maximum ash content was obtained in the cookie with 20% of GP flour. The high ash content in GP (Table 2) is characterized by potassium, calcium, magnesium, sodium, iron, manganese, copper, and zinc, which shows that the addition of GP flour can nutritionally enrich cookies

#### 3.1.4. Pasta

Gaita et al. [35] made pasta, adding 3, 6, and 9% ‘Pinot Noir’ and ‘Riesling’ GP flour. The authors realized that pasta suplemented with GP flour effectively increased the level of phenolic compounds. The incorporation of GP flour of 3, 6, and 9% ‘Pinot Noir’ cultivar resulted in an increase in the total phenolic content of 31, 64%, and 98% in relation to the control. With the incorporation of GP flour of the ‘Riesling’ cultivar, it presented an increase of 39%, 82%, and 125%, respectively. According to the authors, the content of phenolic compounds in the samples depends on the grape variety, harvest time, maceration duration, the pressing conditions used to separate grape juice from the pomace, the pomace drying temperature, and the method of polyphenolic compound extraction.

Soto, Brown, and Ross [32] evaluated noodles and found that the antioxidant activity was higher in noodles containing 20% ‘Cabernet Sauvignon’ grape seed flour when compared to the ‘Merlot’ cultivar.

For Sant’Anna et al. [33], the results showed that the incorporation of 25 g/kg of GP flour increased the polyphenolic concentration and the antioxidant activity in fettuccini. Marinelli et al. [28] noted that spaghetti enriched with GP flour was characterized by a higher content of phenolic compounds, flavonoids, and antioxidant activity.

Marinelli et al. [34], observed that spaghetti enriched with GP flour was characterized by a higher content of phenolic compounds, flavonoids, and, consequently, antioxidant activity when compared to the control. In addition to beneficial effects on human health, the phenolic compounds in grapes are responsible for important characteristics in wines, such as color, astringency, and flavor [39].

### 3.2. Technological Characteristics with Grape Pomace

When flours with higher fiber or protein content are used as a replacement for wheat flour, technological limitations may occur, usually due to increased water absorption, which can make the product dry and brittle, with little elasticity, among other things. Furthermore, in wheat flour, commonly used in bakery products, there is the presence of proteins (glutenins and prolamines) that can form a gluten network, improving flexibility and elasticity [40].

Technological changes can interfere with sensory characteristics, causing lower degrees of acceptance [29]. Such characteristics, when adjusted, add value to the final product, becoming an important alternative to produce healthier foods [25].

#### 3.2.1. Breads

Mildner-Szkudlarz et al. [15] observed that the texture analysis of artisan breads showed significant firmness and gumminess increase with the increment of GP flour. Firmness is the strength needed to bite, while gumminess is the density that persists during chewing [29,32]. However, cohesion and resiliency did not change significantly up to 6% of added GP flour, and chewability up to 4%, while elasticity did not change up to 8% [15].

Hoye and Ross [16] observed that the substitution of more than 5% grape seed flour decreased brightness and volume in breads, with an increase in the firmness and porosity. Sporin et al. [19] also obtained similar results regarding firmness and porosity; they associated this fact to the lower use of wheat flour and a decrease in the amount of gluten. The addition of ‘Merlot’ GP flour had a greater negative impact on bread firmness than ‘Zelen’ GP flour. The textural properties of baked products were affected due to the addition of pomace, possibly caused by a decrease in gluten [17]. These same authors observed that the crust and crumb color of breads changed with the addition of flour, making them darker.

Meral and Dogan [18] evaluated the addition of grape seed flour to replace wheat flour (2.5, 5, 7.5%). This addition improved the rheological properties, increasing the development time and stability of the dough.

#### 3.2.2. Cakes and Muffins

Nakov et al. [21] observed that the width of the cakes did not differ significantly among samples up to 8% of ‘Muscatel’ GP flour addition, and it decreased slightly only in the cakes enriched with 10%. The cake thickness decreased with the gradual increase in GP flour. The volume did not change significantly from 0 to 6% of GP, but drastically decreased thereafter with 10%. This addition increased firmness and chewiness, which may have been caused by a slight reduction in elasticity and cohesiveness. This fact can be attributed to the influence of fibers that interfere in the structure of the dough and can reduce CO_2_ retention [30].

Ortega-Heras et al. [20] observed an increased firmness and chewability of muffins, while the parameters of elasticity, cohesion, resilience, and color decreased.

Bender et al. [6] observed that the flour substitutions affected the firmness, cohesiveness, chewiness, and elasticity of the muffins. Muffins’ firmness formulated with ‘Riesling’ and ‘Tannat‘ grape skin flour increased as the percentage of the grape skin flour increased. The control had the lowest firmness, while the ‘Tannat’ 10% muffin had the highest firmness found.

All samples tested showed greater chewability when compared to the control. This can be explained by the increase in fibers levels, which also increased firmness and chewiness. Yalcin, Ozdal, and Gok [22] also found, in muffins using grape seed flour, that all samples had low elasticity values, commonly found in formulations containing sugar and fat [20]. However, the only formulation that did not have a different elasticity value from the control was the formulation with 5% ‘Riesling’ grape flour [6].

The parameters observed for color varied with the addition of GP flour. The samples enriched with 4%, 6%, and 8% had similar luminosity, while the cake with the addition of 10% was the darkest. A progressive decrease in redness was observed from the control to cakes enriched with 10% GP. In contrast, the more yellowish hue was similar in the control and the cake with 4%, and increased in the other three samples [16]. Bender et al. [6] noticed that the inclusion of grape skin flour decreased the brightness values of crumbs and crusts in muffins. Brightness was more noticeable in formulations with ‘Tannat’ grape skin flour than in formulations with ‘Riesling’ grape skin flour due to the difference in color of the raw materials. The greatest difference in color was observed in muffins made with this grape skin flour, which was attributed to the dark purple color of the grape variety. All muffin crusts were darker than the crumbs, as they were baked, and it is precisely during baking that the Maillard reactions occur [6]. Color is a very relevant property that determines the approval of the bakery products market, because it is directly related to the ingredients used, Maillard reactions, and caramelization of sugar when baked [41].

#### 3.2.3. Biscuits and Cookies

Abreu et al. [28] evaluated cookies and found brightness values from red to yellow, indicating that the grapes were darker. Grape skin color is determined by the amount and composition of anthocyanins [42]. Anthocyanins are pigments responsible for most of the colors in vegetables, with shades between red and blue, which arouse industrial interest for their coloring potential and their antioxidant properties [43].

Poiani and Montanuci [29] observed that, for color, there were interferences, with the cookies tending to be darker as the grape flour was added. When evaluating the density, a significant difference could be observed for the formulation with 6% GP, which was characterized as less dense.

For Samohvalova et al. [27], adding 15% grape seed flour to butter cookies improved specific volume and moistening capacity. Karnopp et al. [25] evaluated formulations with replacement of GP and whole wheat flour and observed that the formulation with 30% of these flours showed the greatest firmness. Abreu et al. [28] found that, as the percentage of flour replacement increased from 5% to 30%, cookies developed fractures and became harder. Cookies are usually less soft and have lower water content when compared to breads, cakes, and pastas. Abreu et al. [28] observed that the higher the percentage of grape flour added, terms such as ‘dry texture’, ‘very dry’, ‘crumbling’ and ‘very dry’ were highlighted in their research.

#### 3.2.4. Pasta

Ungureanu-Iuga, Dimian, and Mironeasa [1] analyzed levels above 3% of grape skin flour and observed greater cooking loss, lower brightness, increased red hue, lower dough elasticity, firmer dough after cooking, and surface roughness. The main cause of cooking losses is gelatinized starch leaching and is more pronounced in gluten-free products due to the absence of the gluten network [32].

For Sant’Anna et al. [33], fettuccini made with ‘Isabel’ GP flour showed that the incorporation of 25 g/kg of GP did not interfere in the quality of the dough, because no differences (*p* > 0.05) were found between the control and 25 g/kg of GP. Marinelli et al. [34] found lower losses of solids in cooking water. Cooking loss is an important parameter in quality assessment because, during pasta cooking, soluble starch parts and other soluble components leach into the water and, as a result, cooking water becomes cloudy and thick [33].

### 3.3. Sensory Characteristics with Grape Pomace

#### 3.3.1. Breads

According to Hayta et al. [17], the sensory evaluation of bread supplemented with 10% GP flour with 14 assessors had the lowest overall acceptance rate, while breads with 2% and 5% addition had similar acceptability values. The same was observed by Mildner-Szkudlarz et al. [15], as the sensory evaluation of the breads with 10 assessors revealed that a maximum of 6% GP can be incorporated to prepare acceptable products.

Sporin et al. [19], using eight trained assessors, realized that the breads’ sensory analysis was associated with a developed sensory profile that included attributes related to texture, aroma, taste, and “mouthfeel”, and a more intense acidic taste and gritty sensation, as well as the stronger aftertaste or off-flavor, were caused by the additions of 15% ‘Merlot’ GP flour.

Hoye and Ross [16], with 87 (panel 1), 97 (panel 2), and 87 (panel 3) assessors, evaluated that breads containing ≥7.5% grape seed flour were characterized by lower consumer acceptance. The authors observed a decrease in general acceptance and bitterness in breads with 10% addition, but noticed a decrease in acceptance of bread astringency and sweetness at 7.5 and 10%.

Because bakery products are widely consumed around the world, it is observed that nowadays there is a growing demand for the elaboration of healthier and alternative bakery products, but these must also maintain the characteristics of good sensory quality [18]. In this sense, the results suggest that pomace can be used in bakery goods, allowing the production of quality products, if they are not added in large percentages [17].

#### 3.3.2. Cakes and Muffins

Nakov et al. [21] observed, with 20 trained assessors, that cakes containing 4% GP had the best sensory quality (Table 2), while cakes with 6% GP had the best texture. The control (0%) was considered pleasant but received slightly lower scores. Thus, the best evaluation was achieved by cakes enriched with 4% grape pomace. The addition of small amounts (4–6%) of grape pomace to food products has been seen to improve nutritional quality and provide better sensory characteristics.

Bender et al. [6] evaluated the differences of two grape cultivars with 51 untrained assessors. There was no significant difference between the sensory attributes of the muffins made with grape skin flour of the ‘Tannat’ variety, but those made with ‘Riesling’ grape skin flour had significant differences (Table 2). It is worth mentioning that ‘Riesling’ grapes are a yellowish-green color, while ‘Tannat’ grapes are purple. The purchase intention for muffins made with ‘Riesling’ skin flour was 5.0 (‘I Would definitely buy’), and for ‘Tannat’ skin flour muffins it was 4.0 (‘Possibly I would buy’). For both cultivars, in grape skin flour muffins the formulations with 5% addition had the highest scores, followed by the 10% formulation.

#### 3.3.3. Biscuits and Cookies

Piovesana et al. [23] proposed the elaboration and evaluation of the acceptability of biscuits made with GP flour and fortified with oat. Three different formulations were tested, 30, 40, and 50%, and were evaluated by 80 untrained assessors. The results showed that overall acceptance, flavor, color, crispness, and purchase intention of the biscuits made with up to 50% of oat flour and GP had significant sensory properties, and the percentages of replacement used in this study were accepted.

Poiani and Montanuci [29] also proposed to develop cookies with grape flour (6, 12, 18%, golden linseed and corn starch). The acceptance test was performed by 100 untrained assessors using a nine-point hedonic scale with terms ranging from “I really disliked it” to “I really liked it”. The 6% formulation had the best results in all attributes, with an overall score of 6.0, with the assessors answering “I slightly liked” the cookies. As the concentration of grape flour increased in the formulations, the color of the cookies became more intense. Therefore, the darker color did not please the assessors, who preferred the lighter colored cookies.

Although Walker et al. [44] found that consumers generally view darker muffins as being healthier and containing more fiber or whole grains, the replacement was not well accepted by the cookie evaluators. For Bender et al. [6], the same was observed, and the higher levels of replacement of grape flour in muffins were not perceived well by consumers.

Abreu et al. [28] characterized the sensory properties of grape skin flour in cookies with 102 untrained assessors. Different formulations were prepared, substituting wheat flour for grape flour (0, 5, 10, 15, 20, and 30%). Cookies with a lower percentage of grape skin flour showed greater acceptance (Table 2).

Karnopp et al. [25], who evaluated the effects of whole wheat (30, 40, 50 g/100 g) and GP flour (20, 25, 30 g/100 g) in cookies observed in sensory evaluation that the addition of GP and whole wheat flour did not affect the acceptance of cookies when compared to the control.

#### 3.3.4. Pasta

Gaita et al. [35] showed that pasta samples with GP skin flour at a level of 3% and 6% had better sensory characteristics when compared to the control. The pasta formulation with the addition of 3% ‘Riesling’ grape skin flour revealed the best sensory properties, followed by samples with 6% ‘Riesling’, 3% ‘Pinot Noir’, and 6% ‘Pinot Noir’. The yellowish-green color of the cultivar ‘Riesling’ did not affect the color of the pasta. This makes it even more interesting to apply grape pomace to products, because, generally, one limitation pointed out in the sensory assessments was the purple color in breads or pasta, which can be considered unusual.

Ungureanu-Iuga and Dimian et al. [1] observed, with nine semi-trained assessors, that the less promising sensory characteristics were associated with bitterness, seed texture, aftertaste, and unpleasant color. Acceptable color, textural, and sensory characteristics were obtained for samples with grape skin levels up to 3% and whey powder up to 15%. Texture is an important attribute in biscuits and cookies, as they are crunchier when compared to breads, cakes, and pastas. Thus, the lower the percentage of GP flour, with or without seeds, regardless of the grape variety being purple or white, the best the sensory acceptance.

No significant differences were found for the attributes of flavor and mouthfeel between pasta with ‘Cabernet Sauvignon’ grape seed flour (20% wheat flour replacement) and ‘Merlot’ grape seed flour (30% wheat flour) in the study by Soto et al. [32]. The global acceptance of pasta made with ‘Cabernet Sauvignon’ grape seed flour (replacement of 25% for wheat flour) was less accepted than the one made with ‘Merlot’ grape seed flour (replacement of 30% wheat flour) [32].

For Sant’Anna et al. [33], sensory analysis suggested that, with 25 g/kg, the formulation had similar acceptance and color changes when compared to traditional fettuccini pasta. The acceptance of the flavor and texture of the pasta did not depend on the concentration of GP added to the formulation.

Marinelli et al. [34], in their study analysis with 15 trained assessors, found no significant differences between experimental samples for sensory analysis.

## 4. Study Limitations

Studies with pasta, bread, biscuits, and cakes were carried out in order to analyze the chemical, technological, and sensory properties of products with GP flour. It is always important to provide most of the details of the ingredients used to develop products. Difficulties were found throughout some papers, as follows: the type of bread that was made or the grape variety used and the parts included in the grape pomace were not always specified in the studies, and papers used different flours as the base, not only wheat. In some papers, the standardization of studies was a factor that influenced the comparison of chemical, technological, and sensory characteristics.

## 5. Conclusions

An improvement in chemical quality, such as fibers, anthocyanins, and micronutrients content was observed with the addition of GP flour. Anthocyanins were present in all samples, due to the composition of the grape skins. However, in relation to the addition of GP flour, technological quality seems to be impaired. Color and texture were quality parameters that showed changes in most studies, both positively and negatively. Thus, despite presenting health benefits, the technological quality still needs to be further studied, so that higher amounts of GP flour can be used.

Samples made with a maximum of 10% of added GP flour seem to be the best accepted. On the other hand, the higher the percentage of GP flour, the healthier benefits they seem to promote. The degree of global acceptability of such products needs to be further studied to find more satisfactory results when compared to conventional pasta and bakery products, such as wheat flour products.

Biscuits and cookies seem to have the best global acceptance. The degree of acceptance still seems to be lower for the use of GP flour when it is included in high quantities in products.

For future perspectives, our research group intends to evaluate the chemical, technological, and sensory quality of grape pomace flour in meat products.

## Figures and Tables

**Table 1 foods-11-03812-t001:** Studies carried out with grape pomace (GP) flour in pasta and bakery products in combination or not with other flours.

Author/Year	Products/ Grape Pomace/ Flour Association	Grape Pomace Flour (%)			
Mildner et al. [15]	**Bread** Grape skin pomace flour Rye flour	4–6	8	10	
Hoye e Ross [16]	**Bread** ‘Merlot’ grape seed flour Wheat flour	2.5	5	7.5	10
Hayta et al. [17]	**Bread** ‘Emir’ grape flour Wheat flour	2	5	10	
Meral e Dogan [18]	**Bread** Grape seed flour Wheat flour	2.5	5	7.5	
Sporin et al. [19]	**Bread** ‘Zelen’ and ‘Merlot’ grape pomace flour	6	10	15	
Bender et al. [6]	**Muffins** ‘Riesling’ and ‘Tannat’ grape skin flour Wheat flour	5	7.5	10	
Ortega-Heras et al. [20]	**Muffins** Red grape pomace flour (without seed) White grape pomace flour (without seed) Whole wheat flour			10 10	20 20
Nakov et al. [21]	**Cake** ‘Muscatel’ grape flour Wheat flour	4–6	8	10	
Yalcin, Ozdal e Gok [22]	**Muffins** Grape seed flour Whole wheat flour Siyez whole wheat flour Oat flour		7.5	15	
Piovesana et al. [23]	**Cookies** Grape seed and skin flour Oat flour		15	20	25
Acun e Gül [24]	**Cookies** Whole grape pomace flour Seedless grape pomace flour Grape seed flour Wheat flour	5 5 5	7.5	10–15 10–15	
Karnopp et al. [25]	**Cookies** ‘Bordeaux’ grape pomace flour Whole wheat flour		20	25	30
Maner, Sharma e Banerjee [26]	**Cookies** ‘Cabernet Sauvignon’ grape pomace flour Wheat flour	5	10–15	20	
Aksoylu, Çagindi e Köse [21]	**Biscuits** Grape seed flour	5			
Samohvalova et al. [27]	**Butter biscuits** Grape seed flour Wheat flour	5	10–15	20	
Abreu et al. [28]	**Cookies ** ‘Bordeaux’ grape skin flour Wheat flour	5	10–15	20	30
Poiani e Montanuci [29]	**Cookies** Grape flour Golden linseed flour Maize starch	6	12	18	
Theagarajan et al. [30]	**Cookies ** ‘Muscatel’ flour Refined Wheat Flour	2	4–6	8	
Sena Júnior, Menezes e Nascimento [31]	**Cookies ** Grape pomace flour Cassava Starch Wheat flour			25	
Soto, Brown e Ross [32]	**Noodles** ‘Cabernet Sauvignon’ grape seed flour ‘Merlot’ grape seed flour Multipurpose flour			20 30	
Sant’Anna et al. [33]	**Fettuccini ** ‘Isabel’ grape pomace flour Wheat flour		25	50	75
Marinelli et al. [34]	**Spaghetti ** Grape pomace flour Durum wheat semolina flour		15		
Gaita et al. [35]	**Pasta ** ‘Pinot Noir’ grape pomace skin flour ‘Riesling’ grape pomace skin flour Wheat flour	3 3	6 6	9 9	
Ungureanu-Iuga, Dimian e Mironeasa [1]	**Rigatoni ** Grape skin flour Whey powder Corn flour Maize starch	1	3	5	

The products already elaborated in previous studies were organized in sequence and shown in bold.

**Table 2 foods-11-03812-t002:** Chemical, technological, and sensory characteristics of products using grape pomace in pasta and baked products.

Product	Authors/ Countries	Chemical	Technological	Sensory
**Bread**	Mildner et al. [15]—Poland	High levels of total fiber, ash, and phenolic compounds.	Increased firmness and gumminess of the breads as the grape flour increases.	Good acceptance with the addition of 4 to 6% grape flour.
**Bread**	Hoye e Ross [16]—USA	Increase in total phenolics with increasing concentration of grape flour.	Increased grape flour decreased the brightness and volume of the bread and increased the porosity and firmness of the bread.	Breads containing ≥7.5% of grape flour had lower acceptance.
**Bread**	Hayta et al. [17]—Turkey	Increased total phenolic content and antioxidant activity according to the addition of grape flour.	Increased color and firmness of breads according to the addition of grape flour.	Moderate acceptance with addition of 2% and 5%. Addition of 10% with the lowest overall acceptance score.
**Bread**	Meral e Dogan [18]—Turkey	Increased antioxidant activity.	Improved stability and extensibility of the dough.	-
**Bread**	Sporin et al. [19]—Slovenia	Increase in total phenolics in cultivar ‘Merlot’ 15%, but without statistically significant difference between cultivars ‘Merlot’ and ‘Zelen’.	‘Zelen’ 15% with less volume. ‘Merlot’ 15% and ‘Zelen’ 6% had an impact on firmness. The 15% ‘Merlot’ stain was the most different from the control.	Good global acceptance, better in breads with the ‘Zelen’ cultivar (due to lesser feeling of sandiness).
**Muffins**	Bender et al. [6]—Brazil	The amount of fiber in the sample with 10% grape flour was double compared to the control.	Increase in the degree of firmness and color as the amount of grape flour increases.	Good acceptance for the attributes: color, flavor, smell, texture, and acceptability.
**Muffins**	Ortega-Heras et al. [20]—Spain	Increased fibre content.	Increased firmness and chewiness. Decrease in elasticity, cohesion, resilience, and color parameters.	High acceptability of muffins that incorporated white and red grape pomace products at 10% concentrations.
**Cake**	Nakov et al. [21]—Bulgaria	Increased content of ash, lipids, proteins, fibers, free phenolic compounds, anthocyanins, and the total content of polyphenols. Decrease in moisture and pH.	It progressively decreased in thickness as the grape flour increased. Decreased volume with 10% grape pomace.	Better sensory quality with 4% ‘Muscatel’ grape flour.
**Muffins**	Yalcin, Ozdal e Gok [22]—Turkey	Increased total phenolic content and antioxidant capabilities with the addition of grape seed meal.	Increase across all muffin samples for firmness, chewiness, crumb and crust color.	There were no significant differences between samples and muffins.
**Cookies**	Piovesana et al. [23]—Brazil	-	Did not differ significantly from each other.	The 50% treatment was less accepted in the flavor attribute.
**Cookies**	Acun e Gül [24]—Turkey	Increased total dietary fiber and total phenolics as increased pomace flour. The total phenolic and antioxidant activity of cookies containing 10% seed meal was considered higher compared to the other samples.	No significant difference.	Better acceptance with 5% seed meal and purchase intent. When the use level of GP flours exceeded 10% for all cookie samples, global acceptance and affordability declined.
**Cookies**	Karnopp et al. [25]—Brazil	Increased fiber content, antioxidant activity, and total phenolic content.	Decrease in water activity. Increased firmness and fracturability of cookies.	No significant differences.
**Cookies**	Maner, Sharma e Banerjee [26]—India	Increased antioxidant properties, total phenolics, flavonoids, and anthocyanins with the addition of wine GP flour.	Increased color intensity as the amount of wine grape pomace flour increases.	The addition of 5% wine GP flour received the highest average.
**Biscuit**	Aksoylu, Çagindi e Köse [38]—Turkey	Increased total phenolic content of cookies.	-	-
**Butter biscuit**	Samohvalova et al. [27]—Ukraine	-	Reduces its tensibility and elasticity increase. Improves the specific volume and wetting ability of cookies with 15% addition.	-
**Cookies**	Abreu et al. [28]—Brazil	Increased fiber content compared to control. High values of phenolic compounds.	-	Good overall acceptance for appearance, aroma, taste, texture, and overall impression.
**Cookies**	Poiani e Montanuci [29]—Brazil	Reduced pH as the grape flour increased.	More browning as the grape flour increased.	Appraisers preferred cookies with 6% addition—the lowest percentage of grape flour.
**Cookies**	Theagarajan et al. [30]—India	Increased protein and fiber content. Higher antioxidant content and lower anthocyanin losses in cookies with 6% GP flour.	Has not affected the technological parameters of cookies.	Best flavor for 4% and 6% Muscatel GP flour.
**Cookies**	Sena Júnior, Menezes e Nascimento [31]—Brazil	Decreased moisture content compared to standard cookies. Lower protein and lipid concentration compared to the standard. Carbohydrate value was higher than sample cookies.	-	-
**Noodles pasta**	Soto, Brown e Ross [32]—USA	Increased antioxidant activities with 20% Cabernet Sauvignon grape seed flour.	-	Samples with 20% of Cabernet Sauvignon grape seed flour had low consumer acceptance.
**Fettuccini pasta**	Sant’Anna et al. [33]—Brazil	Increased polyphenol concentration and antioxidant activity with incorporation of 25 g/kg of GP flour.	The incorporation of 25 g/kg of GP flour did not affect the quality of the dough cooking.	The acceptance of the flavor and texture of the dough was not related to the amount of GP flour.
**Spaghetti pasta**	Marinelli et al. [34]—Italy	Increased total phenolic and flavonoid content and greater antioxidant activity.	Low cooking losses.	No significant difference between samples.
**Pasta**	Gaita et al. [35]—Romania	Increased total polyphenol content and antioxidant capacity.	-	Best acceptance with 3% ‘Riesling’, followed by samples with 6% ‘Riesling’, 3% ‘Pinot Noir’, and 6% ‘Pinot Noir’.
**Rigatoni pasta**	Ungureanu-Iuga, Dimian e Mironeasa [1]—Romania	-	Increased cooking loss to levels bigger than 3% of grape skin flour, reduced brightness, increased red nuance, less dough elasticity, firmer cooked dough, rough surface.	Acceptable sensory characteristics for samples with up to 3% grape skin flour and powdered whey up to 15%. Decreased acceptance in samples above 3%. They were associated with bitterness, seed texture in the mouth, aftertaste, and unpleasant color.

The products already elaborated in previous studies were organized in sequence and shown in bold.

## Data Availability

Not applicable.

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
