# Peer review of "Chemical, Technological, and Sensory Quality of Pasta and Bakery Products Made with the Addition of Grape Pomace Flour"

_foods, 2022, doi:10.3390/foods11233812_

Round 1
Reviewer 1 Report
The review analyzes the scientific publications concerning the use of grape pomace added to foods such as pasta and baked goods. The analysis focuses on the qualitative, sensory and technological effects. However, the well-written article would need improvements to be publishable.
First of all it is bit poor in articles to be considered a review, so perhaps it would be more correct to name it mini review.
The introduction is somewhat short and should be completed by adding the scope of the work and some bibliographical references and similar revision work.
The paragraph corresponding to lines 51-60 should be moved to materials and methods, and replaced by updated references as indicated in the previous comment.
Author Response
Dear Editor and Referee,
We would like to thank the referees and the editor for your valuable time during the review process and the insightful comments which helped us to improve the quality of our manuscript. We have appreciated all the considerations about our paper and we agreed with all of them. We have added the suggestions and corrections pointed by the referees. Questions and suggestions are in black and answers in blue. If something is not exactly as it was suggested, please let us know and we can write it again.
Referee(s)' Comments to Author:
Reviewer: 1
The review analyzes the scientific publications concerning the use of grape pomace added to foods such as pasta and baked goods. The analysis focuses on the qualitative, sensory and technological effects. However, the well-written article would need improvements to be publishable.
Response: We thank the referee for the gentle and encouragement words about our paper. Everything that the referees requested, we added, and we agree that this version is much more complete with these suggestions.
First of all it is bit poor in articles to be considered a review, so perhaps it would be more correct to name it mini review.
Response: The paper has increased in length more 4 pages and we have added 11 references. We looked at the Instructions for Authors on the website: https://www.mdpi.com/journal/foods/instructions , but the journal does not subdivide it into "Mini review" nor does it mention the minimum number of pages or number of characters, we believe they only work with the term "Review".
The introduction is somewhat short and should be completed by adding the scope of the work and some bibliographical references and similar revision work.
Response: We have added 6 new paragraphs in the Introduction, we hope you like the new version of our paper, and we wish that with these new approaches included, the gap you felt previously has been solved.
The paragraph corresponding to lines 51-60 should be moved to materials and methods, and replaced by updated references as indicated in the previous comment.
Response: It was corrected. Thank you for the suggestion. Please, see L.82-98
We are very grateful,
The authors

Reviewer 2 Report
- Please rewrite and organize the abstract according to the following context:
A short introduction, hypothesis (aim) of the study, methods, the most important quantitative results, a general conclusion, and future prospective
-The scientific name should be in italic, please check
- Please state the aim of the study well at the end of the introduction section
- A subsection about the biological activities of grape pomace should be added. It is very important for readers to add that before talking about the chemical characteristics and technological parts
-The design of the manuscript is confusing and should be rewritten with good form.
- Table 1. is confusing, please reformat it
- Conclusions section, please highlight the future standpoint well.
- Manuscript has grammatical errors, please check.
Author Response
Dear Editor and Referee,
We would like to thank the referees and the editor for your valuable time during the review process and the insightful comments which helped us to improve the quality of our manuscript. We have appreciated all the considerations about our paper and we agreed with all of them. We have added the suggestions and corrections pointed by the referees. Questions and suggestions are in black and answers in blue. If something is not exactly as it was suggested, please let us know and we can write it again.
Referee(s)' Comments to Author:
Reviewer: 2
Please rewrite and organize the abstract according to the following context:
A short introduction, hypothesis (aim) of the study, methods, the most important quantitative results, a general conclusion, and future prospective
Response: It was corrected as suggested by our referee. We added the items that our reviewer requested. The "Future perspectives" that we only placed in the conclusion of the paper, since the abstract would be longer than recommended.
-The scientific name should be in italic, please check
Response: We apologize for our lack of attention; we have tried to correct all of them.
- Please state the aim of the study well at the end of the introduction section
Response: It was rephrased in the paper, we hope you agree with this new version. Certainly, this information is important and if you felt a gap, something was missing. Please, see lines 77-79.
- A subsection about the biological activities of grape pomace should be added. It is very important for readers to add that before talking about the chemical characteristics and technological parts
Response: Thank you for the suggestion, it is a very good one buy the way. Please, see lines: L.59- 64
-The design of the manuscript is confusing and should be rewritten with good form.
Response: We added more information as suggested by our referee. We reorganized and hope it is clearer now.
- Table 1. is confusing, please reformat it
Response: We strongly agree. Thanks for your suggestion. We reorganized and hope it is clearer now.
- Conclusions section, please highlight the future standpoint well.
Response: We added more information as suggested by our referee. Please, see lines: 452-470
- Manuscript has grammatical errors, please check.
Response: We apologize for our mistakes; we have tried to correct all of them. We have submitted the paper to a native speaker for reading and proofreading.
We are very grateful,
The authors

Reviewer 3 Report
This manuscript investigate in the literature bakery products and pasta made with different degrees of substitution from different flours from grapes. Suggestions for modification are as follows:
Line 31-32, “the mixture of flours from other species, considered promising in the baking”. Could the author introduce “other species”in detail?
Line 34, Could the author introduce Pasta in detail?
Line 44, How to make GP flour? What are the nutritional components and content of GP flour?
Line 48, What is the difference between the grape pomace flour, grape skin flour and grape seed flour?
Line 85, section 2.1.2. Breads. Besides the increase of phenolic acid and fiber composition, are there any other changes? For example, Texture? Flavor? Shelflife?
Line 107, section 2.1.3. Cakes and muffins. What is the difference between GP flour and grape seed flour? What is the difference between “Tannat’ grape flour and “Muscatel” GP flour
Line 163, Could the author explain why the firmness and gumminess increased with the increment of GP flour?
Line 223-225, Could the author explain why the formulation with 30% of these flours showed the greatest firmness? And the percentage of flour replacement increased from 5% to 30%, cookies were likely to develop fractures and become harder?
Line 232-233, Could the author explain why the incorporation of 25g/kg of GP did not interfere in the quality of the dough?
Line 267-268, What is the difference between “ Tannat” and “Riesling”. Why do they have different influences on cakes or muffin?
Line 283-285, Could the author explain why as the concentration of grape flour increased in the formulations, the colour of the cookies became more intense?
Line 300-301, Could the author explain whythe pasta formulation with the addition of 3% of ‘Riesling’ grape skin flour revealed the best sensory properties?
Author Response
Dear Editor and Referee,
We would like to thank the referees and the editor for your valuable time during the review process and the insightful comments which helped us to improve the quality of our manuscript. We have appreciated all the considerations about our paper and we agreed with all of them. We have added the suggestions and corrections pointed by the referees. Questions and suggestions are in black and answers in blue. If something is not exactly as it was suggested, please let us know and we can write it again.
Referee(s)' Comments to Author:
Reviewer: 3
This manuscript investigate in the literature bakery products and pasta made with different degrees of substitution from different flours from grapes. Suggestions for modification are as follows:
Response: We thank our referee for the helpful suggestions. Everything that our referee requested, we added, and we agree that this version is more complete with these corrections.
Line 31-32, “the mixture of flours from other species, considered promising in the baking”. Could the author introduce “other species” in detail?
Response: We added your requests. We hope it is better in this new version of the paper. (Please, see lines: 36-42)
Line 34, Could the author introduce Pasta in detail?
Response: We added the information as suggested by our referee (Please, see lines:46- 50)
Line 44, How to make GP flour? What are the nutritional components and content of GP flour?
Response: We included the information as suggested by our referee Please, see lines:65- 70)
Line 48, What is the difference between the grape pomace flour, grape skin flour and grape seed flour?
Response: We hope it is clearer in this new version of the paper. Grape skin and grape seed they are part of pomace. Pomace is the solid remains of grapes, olives, or other fruit after pressing for juice or oil. It contains the skins, pulp, seeds, and stems of the fruit.
Line 85, section 2.1.2. Breads. Besides the increase of phenolic acid and fiber composition, are there any other changes? For example, Texture? Flavor? Shelflife?
Response: We added the information as suggested by our referee (Please, see lines: 129-134; L.137- 141; L.242- 150; L.349-353)
Line 107, section 2.1.3. Cakes and muffins. What is the difference between GP flour and grape seed flour? What is the difference between “Tannat’ grape flour and “Muscatel” GP flour
Response: We added the information as suggested by our referee. Grape pomace generally has more parts of grape, while seed flour, only seed. (Please, see lines: L.274-280)
Line 163, Could the author explain why the firmness and gumminess increased with the increment of GP flour?
Response: We added the information as suggested by our referee (Please, see lines 255- 259)
Line 223-225, Could the author explain why the formulation with 30% of these flours showed the greatest firmness? And the percentage of flour replacement increased from 5% to 30%, cookies were likely to develop fractures and become harder?
Response: We added the information as suggested by our referee (Please, see lines 325-328)
Line 232-233, Could the author explain why the incorporation of 25g/kg of GP did not interfere in the quality of the dough?
Response: We added the information as suggested by our referee (Please, see lines 335-337)
Line 267-268, What is the difference between “Tannat” and “Riesling”. Why do they have different influences on cakes or muffin?
Response: We added the information as suggested by our referee (Please, see lines 374-375). ‘Riesling’- It is a white variety with medium-sized bunches and small, delicate berries, and a yellowish-green color. Its pronounced acidity is its main characteristic, providing intense flavors and, most importantly, longevity, which hardly occurs with most white wines. While ‘Tannat’ grapes are purple, have medium size, are compact and cylindrical and slightly elongated. The highlight is the concentration of color in its skin and also in the fruit pulp, justifying the intense purple color that characterizes the wines made with it.
Line 283-285, Could the author explain why as the concentration of grape flour increased in the formulations, the colour of the cookies became more intense?
Response: We added the information as suggested by our referee (Please, see lines 394-398).
Line 300-301, Could the author explain why the pasta formulation with the addition of 3% of ‘Riesling’ grape skin flour revealed the best sensory properties?
Response: We added the information as suggested by our referee (Please, see lines 416-420).
It is worth mentioning that 'Riesling' grapes are yellowish-green color, while Tannat grapes are purple. The yellowish-green color of the cultivar 'Riesling' did not affect the color of the paste. It makes it even more interesting to apply grape pomace in products, since one of the limitations pointed out in the sensory panel was exactly the purple color in breads or pasta, for example, which can be considered unusual.
We are very grateful,
The authors

Round 2
Reviewer 2 Report
The MS has been significantly improved and I appreciate the authors' efforts to respond to reviewer comments.
Reviewer 3 Report
The revised manuscript is improved, and my concerns have been addressed.